# Nano Serpentine Powders as Lubricant Additive: Tribological Behaviors and Self-Repairing Performance on Worn Surface

**DOI:** 10.3390/nano10050922

**Published:** 2020-05-09

**Authors:** Binbin Wang, Zhaodong Zhong, Han Qiu, Dexin Chen, Wei Li, Shuangjian Li, Xiaohui Tu

**Affiliations:** Institute of Advanced Wear & Corrosion Resistance and Functional Materials, Jinan University, Guangzhou 510000, China; wangbinbin911129@163.com (B.W.); zhongzhaodong@hotmail.com (Z.Z.); hanqiu@stu2018.jnu.edu.cn (H.Q.); dxchen@jnu.edu.cn (D.C.); liweijnu@126.com (W.L.)

**Keywords:** serpentine, lubricant additive, friction and wear, self-repairing

## Abstract

Natural serpentine powders are applicable as additives for various lubricating oils. However, no uniform theories explain their tribological performance, lubrication, and wear mechanism, especially their self-repairing mechanism. Herein, the influence of different nano serpentine powders (NSPs) contents in liquid paraffin on the friction and wear characteristics of steel balls and the self-repairing process of NSPs on the worn surface were studied. Results show that the optimal amount of NSPs was 0.5 wt %. Relative to those of the base oil, the friction coefficients and wear spot diameters were reduced by 22.8% and 34.2%, respectively. Moreover, the long-term tribological test shows that the wear scar diameter decreased slightly after 3 h, reaching the state of dynamic balance between wear and repair. The outstanding tribological performance should be attributed to the formed bilayer tribofilm, the first layer of which contains nanoparticles surrounded by lubricants and the second layer of which contains nanoparticles compacted onto the surface of the steel ball.

## 1. Introduction

Lowering friction coefficients and improving the wear-resistance of metals is of great significance for improving the reliability, efficiency, and service life of moving mechanical equipment [1]. Introducing lubricating oil/grease is one of the most effective ways to establish tribo-systems with antifriction and wear-resistant properties [2]. The application of additives could improve the physical and chemical performances of lubricating oils, by either endowing them with new features or strengthening their original performance to meet higher technical requirements. Therefore, additives are the important components of lubricating oils, including high-end modern lubricating oils. Traditional additives, such as sulfonate [3], sulfide alkylphenols [4], and dialkyl dithiophosphate [5], are widely used as extreme pressure anti-wear additives, but they may cause serious environmental pollution. Borate is another type of extreme pressure anti-wear additive with excellent performance [6,7]. It has excellent oxidation stability, antirust property, and outstanding sealing performance. However, its storage stability, hydrolysis stability, and demulsification ability need to be further improved. Therefore, there is urgent demand to explore novel and environmentally friendly additives with excellent tribological performance in applied science and engineering fields.

With outstanding anti-friction property, wear-resistance performance, insensitivity to temperature, and self-repairing function, micro/nano-scaled additives, such as graphene [8], MoS_2_ [9], ZrS_2_ [10], silicate [11], and two-dimensional (2D) Ti_3_C_2_T_x_ nanosheets [12,13] have been highly focused recently. Among various additives, ultrafine natural powders have attracted much attention due to their low cost, facile preparation procedure and good tribological performance. As a typical natural mineral powder, serpentine mineral (Mg_6_(Si_4_O_10_)(OH)_8_) has a representative schistose structure composed of Si–O and Mg–O/OH sheets. The interlaminar forces are the hydrogen bond and van der Waals force. Serpentine mineral is easily dissociated under extreme pressure, because of its weak molecular structure. Hence, it endows lubricating oils with excellent anti-friction and wear resistance performance, and improves the mechanical properties of Fe-based frication pairs [14].

However, the current research on serpentine powders as a lubricant additive has two drawbacks. On the one hand, the experimental results and wear mechanisms lack conformity. Zhang et al. [15,16] found that the optimal amount of serpentine powders was 0.5 wt %. They argued that the tribofilm contained valuable graphite and organic compounds. Yu et al. [14,17] insisted that the addition of 1.5 wt % serpentine to oil was the most efficient in reducing friction and wear, and that the tribological film has no graphite or organic compounds. However, in the later experiments, they used the surface-modified serpentine powders with an average size of 1.0 μm and found that a nanocrystalline tribofilm that is mainly composed of Fe_3_O_4_ FeSi, SiO_2_, AlFe, and Fe-C compound (Fe_3_C), with a thickness of 500–600 nm [18]. By contrast, Qi et al. [19,20] believed that isomorphic replacement and the tribochemical reaction between magnesium silicate and the metal matrix explains the excellent tribological performance of serpentine. On the other hands, although self-repair capability of serpentine is widely accepted, previous research mainly focused on the rough morphologies [19], elemental compositions [21], or mechanical properties [22] of self-repairing layers formed in a certain sliding time. The self-repairing performance of serpentine in different friction periods and the nanoscaled morphologies of self-repairing films were rarely studied. However, those are crucial for understanding the self-healing mechanism of serpentine and improving its tribological performance.

Bearing the above perspective in mind, we undertook the current study by investigating the tribological properties of different mass fractions of serpentine additives in liquid paraffin by using a standard four ball friction and wear tester. The influence of serpentine ultrafine powders on the worn surface in different periods was studied to reveal its named self-healing ability. The antifriction, wear-resistance, and self-healing properties of natural serpentine powders were further explored to provide guidance on their application as additives in modern high-end lubricating oils.

## 2. Experiment Methods

### 2.1. Preparation of Lubricants

The raw material is an artificially crushed natural serpentine ore with a diameter of 0.1~0.5 mm. It was milled by high-energy ball milling at 260 r/min for 12 h, to obtain nanoscaled serpentine powder (NSPs). The base oil for the test is liquid paraffin (LP, Tianjin Damao chemical reagent company, Tianjin, China; Analytical pure).

LP + NSPs lubricants were obtained using the planetary ball milling method and oleic acid was used to modify the surface of the serpentine powders to ensure good dispersibility. Mixtures with the mass fractions of NSPs of 0%, 0.1%, 0.25%, 0.5%, and 1.0% were compounded using the planetary ball milling method at a speed 250 r/min for 90 min. The oil samples were further dispersed by ultrasonic cleaner for 2 h, and lubricants with different mass fractions of NSPs were obtained.

### 2.2. Characterizations of NSPs and Worn Surface

Powder microstructures were observed using a transmission electron microscope (TEM, FEI tecnai G2 TF20, FEI Corporation, Hillsboro, Oregon, USA; accelerating voltage of 200 KV). Particle size distribution of the NSPs was determined via a laser diffraction analyzer (Bettersize2000LD, Dandong Baxter instrument Co. Ltd, Dandong, Liaoning, China). The phase compositions of the powders were determined using a Philips X’Pert MPD X-ray diffractometer (XRD, X’Pert MPD, Royal Philips Corporation, Amsterdam, Netherlands; Cu-Kα radiation, potential 40 KV, current 150 mA) over a 2θ range of 5–60°.

After the friction and wear test, the friction pairs were ultrasonically cleaned in anhydrous ethanol for 30 min. The worn surface of the friction pairs was observed using TEM. The thin foil of the worn surface of the friction pairs used for TEM observation was obtained via a focused ion beam (FIB, Helios nanolab 600, FEI Corporation, Hillsboro, Oregon, USA). The worn surface of the friction pairs was analyzed using a scanning electron microscope (SEM; JSM-5600LV, JEOL Corporation, Akashima, Tokyo, Japan, equipped with an energy dispersive spectrometer (EDS)). The local chemical structure of the worn surfaces of the frictional balls was determined by a Jobin Yvon HR800 Raman spectrometer (Raman spectrometer, Jobin Yvon HR800, Horiba Ltd, Paris, France).

### 2.3. Tribological Test

The friction and wear behaviors of the NSPs addictive are explored using a four-ball friction tester (MS-10a, Xiamen Tianji Automation Co., Ltd., Xiamen, Fujian, China). GCr15 bearing steel balls (composition: 0.95%–1.05% C, 0.25%–0.45% Mn, 0.15%–0.35% Si, and 1.40%–1.65% Cr) with a diameter of 12.7 mm and roughness (Ra) of 20 ± 2.3 nm were used as friction pairs. Before the friction test, the steel balls were ultrasonically cleaned in anhydrous ethanol for 30 min to remove surface contaminants. The experimental load, rotating speed and temperature were 392 N (corresponding to the maximum Hertz pressure of 2293 MPa), 1250 rpm, and 25 ± 2 °C, respectively. The coefficient of friction (COF) was dynamically recorded in a computer. The average spot diameters (AWSDs) were an Olympus microscope with an accuracy of 0.01 mm. Five repeated experiments were performed to eliminate the occasionality of the reported results.

## 3. Results and Discussion

### 3.1. The Characterization of Powders

The TEM image, XRD pattern, and size distribution of the milled serpentine powders are shown in Figure 1. The TEM image in Figure 1a shows that a large proportion of the treated natural serpentine powders via high-energy ball milling are irregularly shaped. The cumulative distribution curve clearly shows that the treated powders follow a typical Gaussian distribution. Specially, the particle sizes spanned from 1.3 nm to 126 nm, d_50_ is 54 nm. Besides, the average diameter of 94% of the serpentine was below 100 nm. The nano-scale of NSPs was favorable for the dispersible stability in LP.

The XRD pattern of the NSPs in Figure 1b shows that the sample had three characteristic diffraction peaks of d = 0.7282, 0.3617, and 0.2519 nm, which can be respectively indexed to the crystal planes of (0 0 1), (1 0 2), and (16 0 1) of antigorite mineral (JCPDS No. 22-1163), which is one kind of the three structures of serpentine. Meanwhile, other diffraction peaks are also presented, suggesting that the milled powders may contain minor amounts of other elements. X-ray fluorescence analysis was conducted to further detect the composition, and the results are listed in Table 1.

### 3.2. Tribological Performance

Figure 2a presents the COF curves of the steel balls as a function of sliding time. Figure 2b shows the AWSD and COF values of the steel balls. Without the addition of NSPs, the COF increased sharply to a relatively steady value of 0.118. The figure also presents that the AWSD was the largest (829.78 μm), indicating the severe wear of the steel balls. These results prove that as a kind of common lubricant, LP plays a certain role in lubrication. However, its lubrication effect was not stable and its lubrication performance was poor.

A distinct scenario is displayed after introducing different contents of NSPs into LP. Figure 2b shows that the AWSD of the steel balls under the lubrication of LP + NSPs was effectively reduced. However, the COFs show different trends given the different amounts of NSPs. COF and AWSD show decreasing trends with the increase of the concentration of milled powders. When the content of NSPs increased to 0.5 wt %, the oil sample exhibited the best tribological performance. Compared with that of the base oil, the average COF decreased from 0.092 to 0.071, and the AWSD of the steel ball decreased from 829.78 to 546.00 μm. However, as the content of NSPs increased to 1.0 wt %, the COF and AWSD presented an increase of a different magnitude. These results show that the optimal concentration of serpentine powders as a lubricant additive was 0.5 wt %. When the concentration was lower or higher than this value, the anti-friction and wear-resistance properties were weakened but remain superior to those of pure LP.

Figure 3 shows the SEM morphologies of the worn surfaces of the friction pairs under the lubrication of LP with and without NSPs. A typical plastic deformation zone with numbers of cavities is evident on the worn surfaces of the steel balls under the lubrication of the base oil. This characteristic suggests severe adhesive wear. Moreover, several furrows in the sliding direction resulting from wear debris are evident in the image and indicate mild abrasive wear. Namely, the friction pairs are dominated by adhesive wear and slight abrasive wear under the lubrication of LP. The abundance of oxygen (Table 2) indicates the oxidation of iron. The real contact of a frictional surface is the point-to-point contact, and the asperity contact spots often produce extremely high flash point temperature. Many previous studies [19,22] have confirmed that the flash temperature could be the most crucial factor affecting the tribochemical reaction. The color contrast of LP before/ and after the friction test (Figure 4) indicates that the wear debris has been translated into base oil. To figure out the composition, the colored oil was centrifuged and cleaned thrice, and element analysis, followed to determine the composition. The results show that the two main components of debris are Fe (38.56%, atomic fraction) and O (40.15%, atomic fraction). The results further confirm that the friction pairs are oxidized during sliding under the lubrication of LP.

The plastic deformation zone became narrow, but the number of furrows increased with the addition of small amounts of NSPs (0.1 wt %) into LP. This result implies that the wear mechanism was dominated by abrasive and adhesive wear. Another noteworthy phenomenon is the appearance of several grey regions on the worn surface their effect and composition will be discussed later. The EDS results (Table 2) led to the conclusion that the oxidation of iron was suppressed to a certain extent. Meanwhile, the elements of NSPs began to appear on the worn surface.

As the additive amount increased to 0.5 wt %, the plastic deformation zone and furrows disappeared on the worn surface and replaced by numbers of grey and dark spots. Meanwhile, the worn surface is the smoothest of all the samples. A high surface roughness would result in the reduction of the actual contact area between the friction pairs, leading to a high load concentrated in a small area, and ultimately raising the COF [23]. Therefore, the smoothest surface should be partly responsible for the reduction of the COF. The EDS results (Table 2) demonstrated that numerous elements of NSPs accumulated on the surface, and the oxygen content continued to decline. As a portion part of O may result from the accumulated NSPs, the oxidization of iron was further suppressed under the lubrication of LP + 0.5 wt % NSPs, this condition is conducive to the improvement in wear-resistance.

Figure 5 presents the Raman spectra of the worn surface of the friction pairs under the lubrication of LP and LP + 0.5 wt % NSPs. The intensity of the characteristic peaks of iron oxide [24] (676 cm^−1^) of the worn surface under the lubrication of LP + 0.5 wt % NSPs is lower than that under the lubrication of LP. This finding verified that the oxidization of Fe is further suppressed under the lubrication of LP + 0.5 wt % NSPs. In addition, characteristic peaks of graphite [25], D band around 1340 cm^−1^ and G band around 1580 cm^−1^, are shown in Figure 5b. The result indicates the generation of graphite film onto the worn surface.

As shown in Figure 3d, plenty of scratches and countless dark spots formed as the amount of additives increased. This result implies that the wear mechanism transforms into abrasive wear. Different from the EDS results of Figure 3c (Table 2), the oxidization of Fe did not deteriorate, and the reserved NSPs elements onto the worn surface were few, although the additive amount was doubled. Superfluous NSPs particles would act as abrasives bringing about abrasive wear.

An outstanding characteristic of the worn surface of the friction pairs under the lubrication of LP with NSPs is the distribution of many grey regions. Their composition was investigated to understand the phenomenon out. Figure 6 presents the elemental maps of the worn surface of the balls under the lubrication of LP + 0.5 wt % NSPs. The results show that the worn surface was mainly composed of Fe, C, O, Si, and Mg. Two notable phenomena are presented in the images. On the one hand, the O element tended to gather along with Si, thereby suggesting that the grey region may be mainly composed of SiO_x_. On the other hand, much Fe, C, and Mg elements present in the edges of grey regions, thereby verifying the existence of the transfer layer formed by a tribochemical reaction.

### 3.3. Self-Repairing Performance of NSPs

The following experiments were designed to further verify the self-repairing property of ultrafine serpentine powders. Friction and wear test lasting 1 h was conducted on the steel balls under the lubrication of LP. We, then extracted the used base oil, and dropped a commensurate amount of oil sample containing 0.5 wt % NSPs. We continued the tribological tests in different friction periods. For comparison, the tribological tests on the steel balls were also performed under the lubrication of LP in the same periods.

Figure 7 shows the COF curves and AWSD of the steel balls. It can be seen from the figure that the 0-c stage (the first hour of the experiment) is running in the stage, and the COF curve shows a gradual upward trend. Afterward, the COF value of the steel balls immersed in pure LP increased to 0.13. The AWSD of the steel balls increased with sliding time. By contrast, the COF curve decreased momentarily, decreased slowly, and finally remained at about 0.05 when the oil sample containing 0.5 wt % NSPs was used for the friction and wear test after point C. Unexpectedly, the AWSD of the steel balls increased slowly first, and reached the maximum value after sliding for 3 h; the increase range was evidently smaller than that under the lubrication of LP. However, the AWSD of the steel balls shows a slight decreasing trend with the sliding time. These experimental results indicate that introducing NSPs into LP could reduce the COF improve wear resistance and establish a certain self-repairing function.

To clarify the nature of wear, we examined the SEM morphologies of the worn surface of friction pairs under the lubrication of pure LP under different times. The image of the worn surface after sliding for 2 h (Figure 8b) shows that the plastic deformation zone diminished, while furrows widened and deepened. This indicates that abrasive wear became the main form of wear of frication pairs under the lubrication of LP. Based on the above analysis, we might conclude that through the long time sliding, the oxidation of Fe became severe. As the friction continuing, the several biggest and deepest furrows shown in Figure 8c imply the steel ball suffered extremely serious abrasive wear.

To determine excellent self-repairing of NSPs, we also examined the SEM morphologies of worn surface of friction pairs under the lubrication of LP + 0.5 wt % NSPs under different times. Compared to the worn surface of friction pairs under the lubrication of LP after 2 h (Figure 8b), that under the lubrication of LP + 0.5 wt % (Figure 8b) exhibited thinner furrows and numbers of dark areas. More importantly, a smoothing trend of the worn surface and increasing trend of the dark areas seemed to happen. Namely, to some extent, the LP + 0.5 wt % NSPs possessed the ability to self-repair with the help of the friction behavior.

The TEM analysis for the cross section of the wear scar of friction pair sliding for 4 h was performed to further investigate the unusual phenomenon. The cross-sectional TEM image of the worn surface shows that two compact layers with a total thickness of 600 nm tightly covered the surface of the steel ball as shown in Figure 9a. In the first layer, numbers of grey patches were surrounded by light-colored parts. Meanwhile, a homogeneous layer with a grey color was tightly connected to the surface of the steel ball. The enlarged image shows numerous nanoparticles at the boundary of the two layers. The unusual bilayer structure of the transfer film was probably responsible for the excellent lubricant properties and ultra-long service life. Thereafter, the transfer film with a bilayer structure was analyzed in detail.

The EDS line distribution of the Fe, C, Si, O, and Mg elements (Figure 10) shows that the light color in the transfer film was mainly composed of C, Si, and O elements and that the dark regions comprised of Fe and O elements. More important, the emergence of Fe and O elements was synchronized, thereby proving the existence of the oxide of Fe. Three notable facts should be identified from the high-resolution transmission electron microscopy (HRTEM) images (Figure 11): First, the HRTEM of part A in Figure 9a confirmed that the crystal structure of the iron oxide was Fe_3_O_4_ and Fe_2_O_3_, which were d-spaces of 0.128 and 0.109 nm and 0.145 and 0.149 nm, respectively. Second, the characteristics of the amorphous component were found in the light-colored regions (B region in Figure 9a). Combining the EDS line distribution and Raman spectra results revealed that the amorphous region was mainly composed of graphite. The amorphous graphite results from the breakage of the carbon chain of the lubricating oil due to mechanical and tribochemical actions and as a response to the long-term sliding. The HRTEM image of part B with the combination of light color and dark color shows that amorphous graphite was surrounded by SiO_2_ nanocrystallites. Third, the HRTEM image of the selected regions (part C) in the second layer indicates that the main constituents of the second layer were Fe_2_O_3_ and Fe_3_O_4_, as well as a small amount of SiO_2_.

Serpentine mineral (Mg_6_(Si_4_O_10_)(OH)_8_) has a representative schistose structure composed of Si–O and Mg–O/OH sheets. The non-bridging oxygen atoms in [SiO_4_] pointed to the same direction and connect with Mg^2+^, and consequently formed Mg–O octahedral layers. The two layers were connected by hydrogen bond and van der Waals force. Owing to the existence of the weak molecular, it is easy to dissociate under extreme pressure, which displays good capability of oxygen release. Under the effect of compressive stress, friction shearing stress, and friction reaction, the crystal of the layered structure of serpentine particles that were suspended in oil would be broken. Then oxygen atoms and oxygen-containing species were released for the bond breakage of Si–O tetrahedron and Mg–O octahedral. However, according to Zhao et al. the high flash-temperature made the ion exchange reaction of iron atoms and Mg^2+^, so the Mg element is rarely detected in the worn surface.

In general, iron oxide can be aptly generated with the combined effects of friction-induced heat and contact pressure during sliding motion. The sliding motion facilitates the removal of the iron oxide layer, leaving the fresh Fe to be oxidized. In worse cases, the formed iron oxide particles would plow the worn surface, further accelerating the damage of the components. Thus, the existence of iron oxide is disadvantageous to the protection of substrates. However, several unusual facts are found in the present work. Certain amounts of Fe_2_O_3_, Fe_3_O_4_, and SiO_2_ with sizes ranging from dozens of nanometers to hundreds of nanometers were distributed in the first tribolayer. Moreover, a large amount of lubricant surrounds the wear particles. This condition ensures lubricant ability and prevents the formed iron oxide particles from contacting the worn surface directly. Under high contact pressure, the accumulated Fe_2_O_3_, Fe_3_O_4_, and product of NSPs are compacted, resulting in a compact layer connected to the surface of the steel ball. This behavior prevents the damage of the first layer caused by excessive oxide particles. According to the above analysis, the formed bilayer, the first one of which contains nanoparticles surrounded by lubricants and the second one of which contains nanoparticles compacted onto the surface of the steel ball, was responsible for the self-repairing performance of NSPs.

## 4. Conclusions

The NSPs powders were successfully fabricated by the high-energy ball milling method. The tribological behaviors and self-repairing performance of NSPs on the worn surface have been investigated in detail. The main conclusions can be drawn as follows:

(1) LP + NSPs lubricants possessed outstanding friction-reducing and wear-resistance properties, and the optimal amount of NSPs in LP was 0.5 wt %. Relative to that of the base oil, the friction coefficients and wear spot diameters were reduced by 22.8% and 34.2%, respectively.

(2) The friction pairs were dominated by adhesive wear and slight abrasive wear under the lubrication of LP. Superfluous additives acted as abrasive particles, rough worn surface, and degrade the lubricant performance of base oil.

(3) Compared to base oil, LP + 0.5 wt % NSPs had longer service life. The wear scar diameter contracting slightly after 3 h, reaching the state of dynamic balance between wear and repair.

(4) The outstanding anti-friction and self-repairing performance of LP + 0.5 wt % NSPs should be attributed to the formed bilayer, structure, the first one of which contains nanoparticles surrounded by lubricants, and the second one of which contains nanoparticles compacted onto the surface of steel ball.

## Figures and Tables

**Figure 1 nanomaterials-10-00922-f001:**
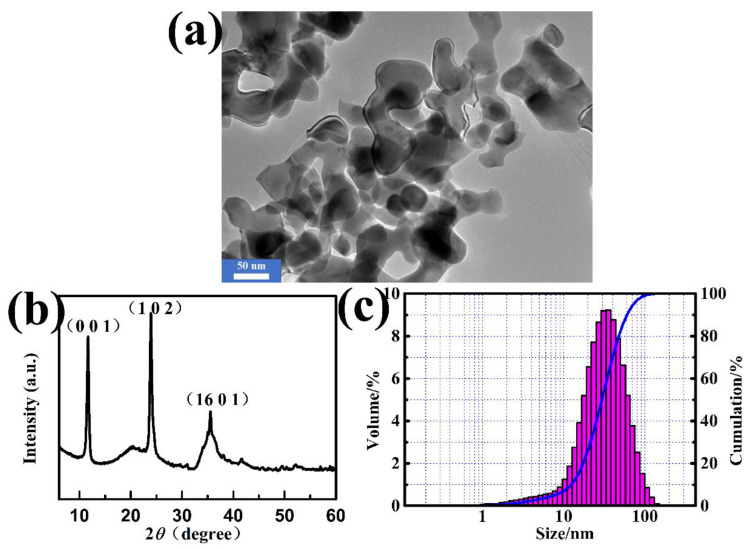
TEM micrograph (**a**), XRD spectrum (**b**), and particle size distribution of milled serpentine powders (**c**).

**Figure 2 nanomaterials-10-00922-f002:**
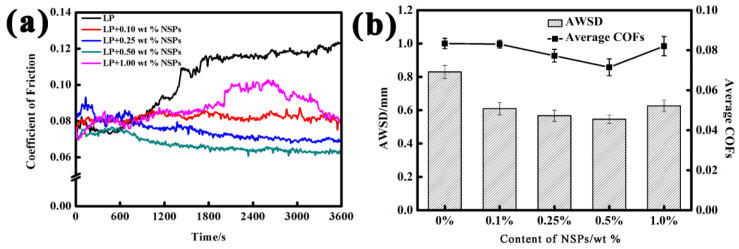
Coefficients of friction (COFs) of samples as a function of sliding time (**a**), and average wear scar diameter and COFs at different contents (**b**).

**Figure 3 nanomaterials-10-00922-f003:**
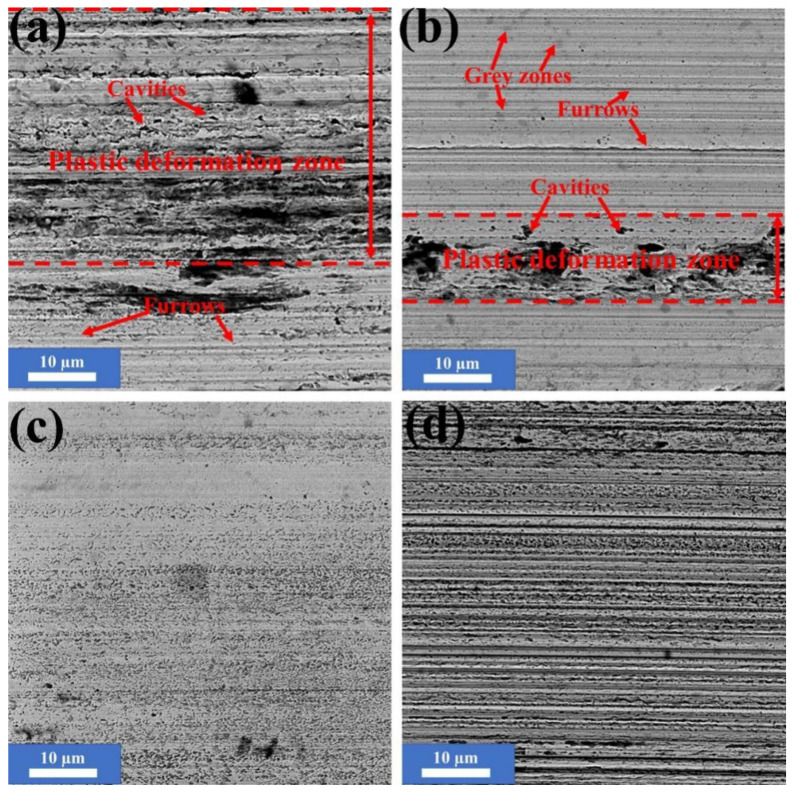
SEM images of worn surfaces under the lubrication of pure liquid paraffin (LP; **a**), LP + 0.1 wt % nanoscaled serpentine powders (NSPs; **b**), LP + 0.5 wt % NSPs (**c**), and LP+1.0 wt % NSPs (**d**).

**Figure 4 nanomaterials-10-00922-f004:**
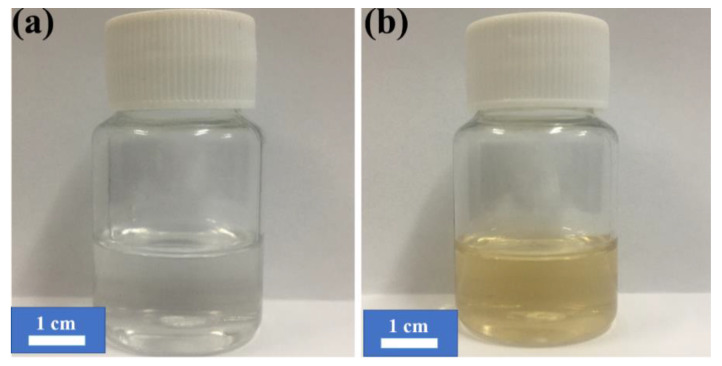
Optical image of LP before (**a**) and after (**b**) the tribological test.

**Figure 5 nanomaterials-10-00922-f005:**
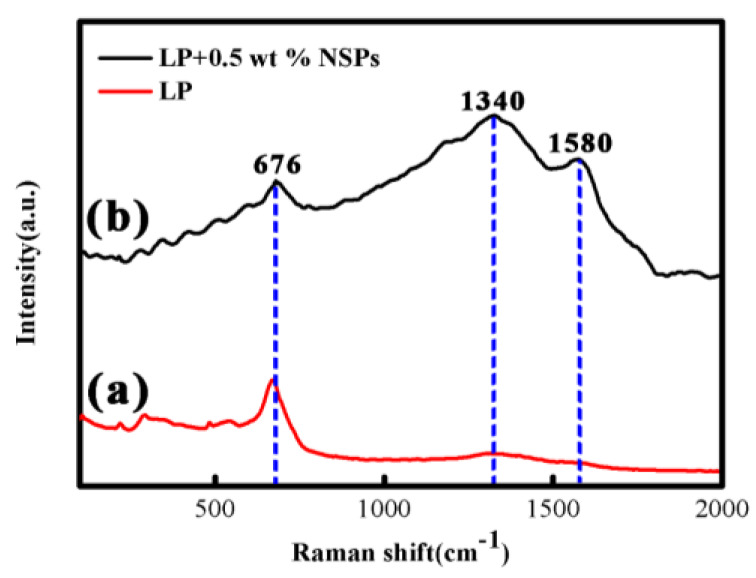
Raman spectra of the worn surface of friction pairs under the lubrication of LP (**a**) and LP + 0.5 wt % NSPs (**b**).

**Figure 6 nanomaterials-10-00922-f006:**
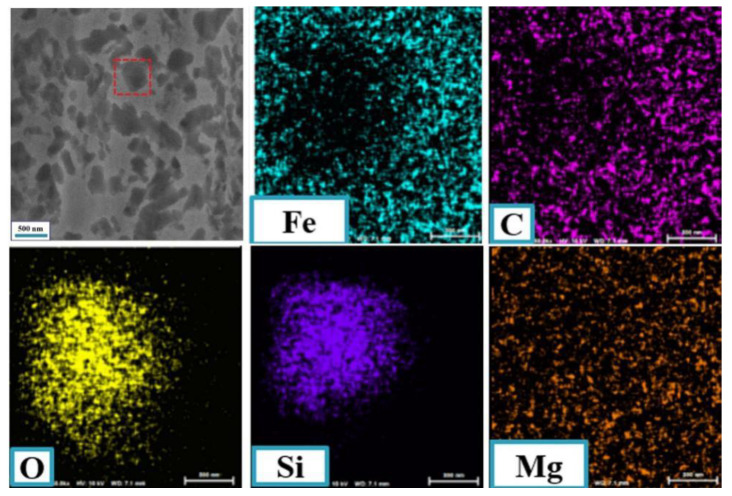
Elemental mapping of the worn surface of the steel ball under the lubrication of LP + 0.5 wt % NSPs.

**Figure 7 nanomaterials-10-00922-f007:**
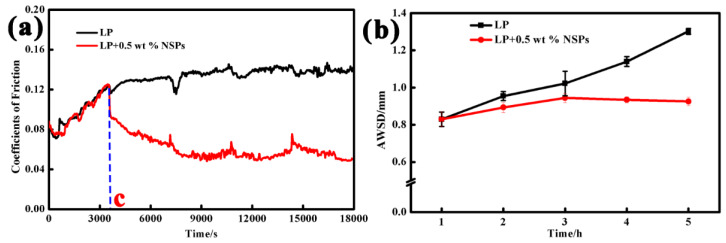
COFs (**a**) and the average spot diameter (AWSD; **b**) of steel balls lubricated with LP + 0.5 wt % NSPs under different times.

**Figure 8 nanomaterials-10-00922-f008:**
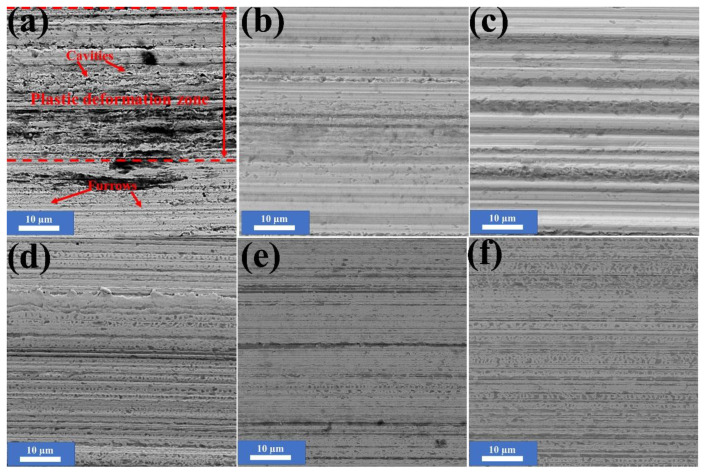
SEM morphologies of worn surface of friction pairs under the lubrication of pure LP ((**a**) 1 h, (**b**) 2 h, and (**c**) 5 h), and LP + 0.5 wt % NSPs ((**d**) 2 h, (**e**) 3 h, and (**f**) 5 h) after different sliding times.

**Figure 9 nanomaterials-10-00922-f009:**
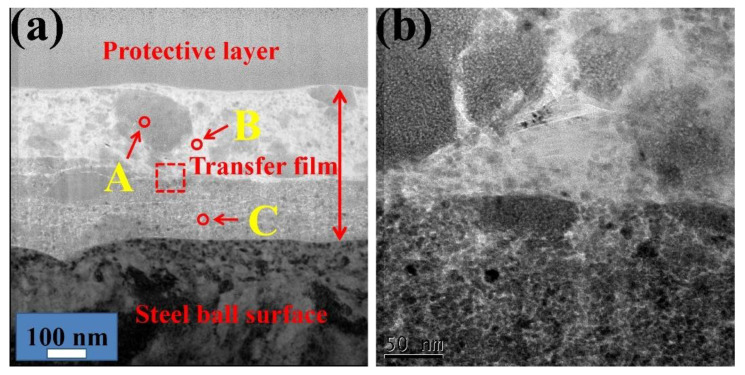
(**a**) Cross-sectional TEM image of the worn surface under the lubrication of LP + 0.5 wt % NSPs after sliding for 4 h and (**b**) enlarged TEM image of red box in (**a**).

**Figure 10 nanomaterials-10-00922-f010:**
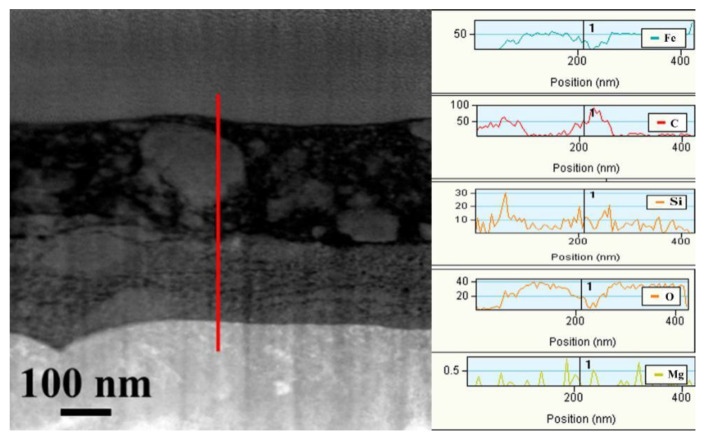
EDS lines distribution of main elements of tribofilm (The color lines in cross-sections corresponding to the direction of EDS lines).

**Figure 11 nanomaterials-10-00922-f011:**
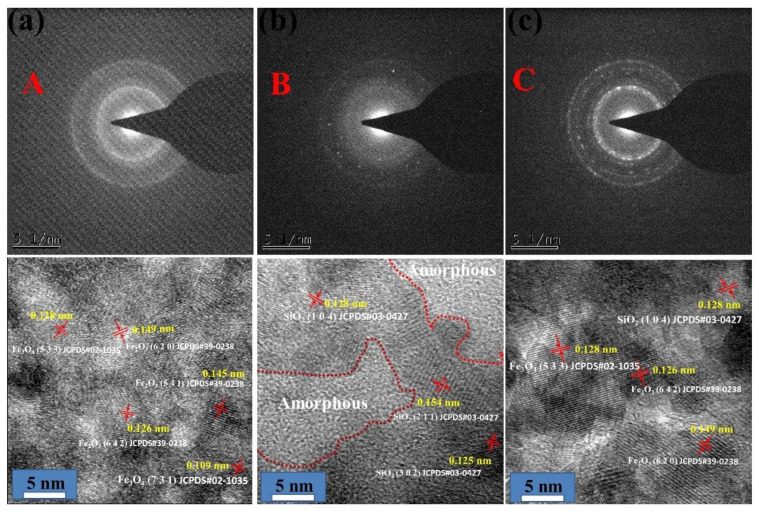
High-resolution TEM image of specified regions in Figure 9a.

**Table 1 nanomaterials-10-00922-t001:** X-ray fluorescence analysis results of the milled serpentine particle.

Oxides	Contents (%)
SiO_2_	47.5
MgO	43.2
Fe_2_O_3_	6.33
Al_2_O_3_	2.97

**Table 2 nanomaterials-10-00922-t002:** The element content (atomic concentration) of the worn surfaces in Figure 3.

Element Area	Fe	O	Si	Mg	Cr
a	57.46	41.08	0.11	0	1.35
b	68.77	28.85	0.58	0.23	1.57
c	72.61	18.34	6.80	0.82	1.43
d	83.63	12.10	2.45	0.41	1.41

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
