# Peer review of "Nano Serpentine Powders as Lubricant Additive: Tribological Behaviors and Self-Repairing Performance on Worn Surface"

_nanomaterials, 2020, doi:10.3390/nano10050922_

Round 1
Reviewer 1 Report
Review of the manuscript "Nano serpentine powders as lubricant additive: Tribological behaviors and self-repairing performance on worn surface" by Binbin Wang et. al.
The authors' work is of interest from the point of view of obtaining additional information and lubricants based on natural serpentine powders and mechanisms providing good tribological characteristics.
At the same time, there are a number of comments on the text of the work.
In fig. 1 shows a setup diagram for tribological tests. As I understand it, the balls rotate freely. The distance from the balls to the walls of the container is very large - it is obvious that upon loading the three lower balls will “leave” to the walls and come out of contact with each other and with the upper fourth ball. Or are crimping rings holding them back?
Section 3 begins with subparagraph 3.3, and not 3.1?
To determine the nature of wear, it was necessary to add studies of scanning microscopy.
The rest of the material of the manuscript is interesting from the point of view of practical application.
Author Response
Thank you for your careful review on our manuscript, and your excellent advices are important to improve the article. A point-by-point response to each of the comments is listed in the attachment. Please see the attachment.

Reviewer 2 Report
- First and foremost, the language of this manuscript needs to be improved. The level of English is not really acceptable.
- Regarding to the use of additives, especially related to new layered materials, the use of them as solid lubricants should be mentioned. Afterwards, it is straighforward to conclude that also mixing them to base oils makes a lot of sense. However, issues related to dispersion stability should be discussed.
- Related to 2d layered materials, newly emerging Mxene nanosheets used as solid lubricant and lubricant additive should be mentioned.
- No stability experiments using zeta potential neither sedimentation analysis or DLS has been shown.
- Please explain why you have chosen the respective tribological testing conditions.
- Figure 1 can be deleted.
- Partially Figure 2 is too small. Also the position and size of a,b and c may be improved.
- Regarding to Figure 3a, have you reached steady state after 3600 s? Why have you selected this short period of time?
- How did you perform the wear analysis to obtain figure 3b? How many measurements did you do?
- The COF evolution of 0.5 % is fairly different in Figure 3 and Figure 8? How do you explain these differences?
- Raman bands should be referenced.
- Conclusions should be shortened.
- Overall, there is a lack of discussion in the entire manuscript. The discussion aspect is by far not enough to justify publication.
Author Response
Thank you very much for your careful review and constructive comments on the manuscript. According to your advice, we have taken the help of a professional agency (EnPapers.com) to proofread the whole manuscript, and the certificate of English editing has also been attached. A point-by-point response to each of the comments is listed in the attachment. Please see the attachment.

Reviewer 3 Report
The article “Nano serpentine powders as lubricant additive: Tribological behaviors and self-repairing performance on worn surface” describes effect of nano-serpentine powders content in liquid paraffin oil on friction and wear of steel surfaces performed on four-balls friction machine. It was revealed that the optimal concentration of nano-particles in oil is 0.5 wt. percentage. The extraordinary tribological and self-repairing properties of the oil with 0.5 wt.% of the powder is featured by the formed bi-layer containing nanoparticles surrounded by lubricants and compacted layer just on the steel surface.
The tribological experiments evidently demonstrated self-repairing ability of the mixed oil. The article surely can be published in Nanomaterials.
For the completeness of the review of the previous research, I would recommend to add in the introduction the article in which the bigger serpentine particles were taken for the study:
Yu, H., Xu, Y., Shi, P., Wang, H., Wei, M., Zhao, K., & Xu, B. (2013). Microstructure, mechanical properties and tribological behavior of tribofilm generated from natural serpentine mineral powders as lubricant additive. Wear, 297(1), 802-810.
Author Response

(The authors gave the same response as above.)

Reviewer 4 Report
Dear authors,
You presented a number of very interesting data, obtained with care and using modern measuring devices and methods. The major conclusion points out the very good performances of LP+NSPs lubricants and establishes the value of 0.5wt% as the optimum percent of NSP in LP+NSPs lubricants.
However in both, introduction and original paper's body you noticed some contradictions for which you presented possible and reliable explanations.
1. Some quantitative values regarding the running conditions used during experiments, as:
- balls' radius, maximum Hertz pressure an sliding velocity,
- evolution of the working temperatures,
-minimum thickness of the elasto-hydrodynamic lubrication film (for the case of rig's lubrication using LP with no NSP),
-the roughness values for balls' surfaces, before testing operation,
would be helpful to evaluate the lubrication and friction regimes in your experiments as well as in those obtainable from neutral literature.
2. To have comparisons data,have you tested NSP powder with a different viscosity fluid ?
Author Response
We are deeply grateful to you for your constructive comments. A point-by-point response to each of the comments is listed in the attachment. Please see the attachment.

Round 2
Reviewer 2 Report
Thanks for addressing muy comment and concerns:
- In my eyes, Figure 1 still needs improvement.
- There are better citations for the ultra-wear resistant self-lubrication properties of Mxenes. You may check the literature.
